# Graph Few-shot Learning with Task-specific Structures

**Song Wang**
University of Virginia
sw3wv@virginia.edu

**Chen Chen**
University of Virginia
zrh6du@virginia.edu

**Jundong Li**
University of Virginia
jundong@virginia.edu

## Abstract

Graph few-shot learning is of great importance among various graph learning tasks. Under the few-shot scenario, models are often required to conduct classification given limited labeled samples. Existing graph few-shot learning methods typically leverage Graph Neural Networks (GNNs) and perform classification across a series of meta-tasks. Nevertheless, these methods generally rely on the original graph (i.e., the graph that the meta-task is sampled from) to learn node representations. Consequently, the graph structure used in each meta-task is identical. Since the class sets are different across meta-tasks, node representations should be learned in a task-specific manner to promote classification performance. Therefore, to adaptively learn node representations across meta-tasks, we propose a novel framework that learns a task-specific structure for each meta-task. To handle the variety of nodes across meta-tasks, we extract relevant nodes and learn task-specific structures based on node influence and mutual information. In this way, we can learn node representations with the task-specific structure tailored for each meta-task. We further conduct extensive experiments on five node classification datasets under both single- and multiple-graph settings to validate the superiority of our framework over the state-of-the-art baselines. Our code is provided at https://github.com/SongW-SW/GLITTER.

## 1 Introduction

Nowadays, graph-structured data is widely used in various real-world applications, such as molecular property prediction [18], knowledge graph completion [47], and recommender systems [45]. More recently, Graph Neural Networks (GNNs) [43, 44, 32, 15] have been proposed to learn node representations via information aggregation based on the given graph structure. Generally, these methods adopt a semi-supervised learning strategy to train models on a graph with abundant labeled samples [19]. However, in practice, it is often difficult to obtain sufficient labeled samples due to the laborious labeling process [10]. Hence, there is a surge of research interests aiming at performing graph learning with limited labeled samples as references, known as *graph few-shot learning* [48, 21, 9].

Among various types of graph few-shot learning tasks, *few-shot node classification* is essential in real-world scenarios, including protein classification [3] and document categorization [30]. To deal with the label deficiency issue in node classification, many recent works [6, 10, 18, 21] incorporate existing few-shot learning frameworks from other domains [33, 27] into GNNs. Specifically, few-shot classification during evaluation is conducted on a specific number of meta-test tasks. Each meta-test

36th Conference on Neural Information Processing Systems (NeurIPS 2022).

task contains a small number of labeled nodes as references (i.e., support nodes) and several unlabeled nodes for classification (i.e., query nodes). To extract transferable knowledge from classes with abundant labeled nodes, the model is trained on a series of meta-training tasks that are sampled from these disjoint classes but share similar structures with meta-test tasks. We refer to meta-training and meta-test tasks as meta-tasks. Note that few-shot node classification can be conducted on a single graph (e.g., a citation network for author classification) or across multiple graphs (e.g., a set of protein-protein interaction networks for protein property predictions). Here each meta-task is sampled from one single graph in both single-graph and multiple-graph settings, since each meta-test task is conducted on one graph. Despite the success of recent studies on few-shot node classification, they mainly learn node representations from the original graph (i.e., the graph that the meta-task is sampled from). However, the original graph can be redundant and uninformative for a specific meta-task as each meta-task only contains a small number of nodes. As a result, the learned node representations are not tailored for the meta-task (i.e., task-specific), which increases the difficulties of few-shot learning. Thus, instead of leveraging the same original graph for all meta-tasks, it is crucial to learn a task-specific structure for each meta-task.

Intuitively, the task-specific structure should contain nodes in the meta-task along with other relevant nodes from the original graph. Moreover, the edge weights among these nodes should also be learned in a task-specific manner. Nevertheless, it remains a daunting problem to learn a task-specific structure for each meta-task due to two challenges: (1) It is non-trivial to select relevant nodes for the task-specific structure. Particularly, this structure should contain nodes that are maximally relevant to the support nodes in the meta-task. Nevertheless, since each meta-task consists of multiple support nodes, it is difficult to select nodes that are relevant to the entire support node set. (2) It is challenging to learn edge weights for the task-specific structure. The task-specific structure should maintain strong correlations for nodes in the same class, so that the learned node representations will be similar. Nonetheless, the support nodes in the same class could be distributed across the original graph, which increases the difficulty of enhancing such correlations for the task-specific structure learning.

To address these challenges, we propose a novel **G**raph few-shot **L**earning framework w**IT**h **T**ask-sp**E**cific st**R**uctures - GLITTER, which aims at effectively learning a task-specific structure for each meta-task in graph few-shot learning. Specifically, to reduce the irrelevant information from the original graph, we propose to select nodes via two strategies according to their overall node influence on support nodes in each meta-task. Moreover, we learn edge weights in the task-specific structure based on node influence within classes and mutual information between query nodes and labels. With the learned task-specific structures, our framework can effectively learn node representations that are tailored for each meta-task. In summary, the main contributions of our framework are as follows: (1) We selectively extract relevant nodes from the original graph and learn a task-specific structure for each meta-task based on node influence and mutual information. (2) The proposed framework can handle graph few-shot learning under both single-graph and multiple-graph settings. Differently, most existing works only focus on the single-graph setting. (3) We conduct extensive experiments on five real-world datasets under single-graph and multiple-graph settings. The superior performance over the state-of-the-art methods further validates the effectiveness of our framework.

## 2 Problem Formulation

Denote the set of input graphs as $\mathcal{G} = \{G_1, \ldots, G_M\}$ (for the single-graph setting, $|\mathcal{G}| = 1$), where $M$ is the number of graphs. Here each graph can be represented as $G = (\mathcal{V}, \mathcal{E}, \mathbf{X})$, where $\mathcal{V}$ is the set of nodes, $\mathcal{E}$ is the set of edges, and $\mathbf{X} \in \mathbb{R}^{|\mathcal{V}| \times d}$ is a feature matrix with the $i$-th row vector ($d$-dimensional) representing the attribute of the $i$-th node. Under the prevalent meta-learning framework, the training process is conducted on a series of meta-training tasks $\{\mathcal{T}_1, \ldots, \mathcal{T}_T\}$, where $T$ is the number of meta-training tasks. More specifically, $\mathcal{T}_i = \{\mathcal{S}_i, \mathcal{Q}_i\}$, where $\mathcal{S}_i$ is the *support set* of $\mathcal{T}_i$ and consists of $K$ labeled nodes for each of $N$ classes (i.e., $|\mathcal{S}_i| = NK$). The corresponding label set of $\mathcal{T}_i$ is $\mathcal{Y}_i$, where $|\mathcal{Y}_i| = N$. $\mathcal{Y}_i$ is sampled from the whole training label set $\mathcal{Y}_{train}$. With $\mathcal{S}_i$ as references, the model is required to classify nodes in the *query set* $\mathcal{Q}_i$, which contains $Q$ unlabeled samples. Note that the actual labels of query nodes are from $\mathcal{Y}_i$. After training, the model will be evaluated on a series of meta-test tasks, which follow a similar setting as meta-training tasks, except that the label set in each meta-test task is sampled from a distinct label set $\mathcal{Y}_{test}$ (i.e., $\mathcal{Y}_{test} \cap \mathcal{Y}_{train} = \emptyset$). It is noteworthy that under the multiple-graph setting, meta-training and meta-test tasks can be sampled from different graphs, while each meta-task is sampled from one single graph.

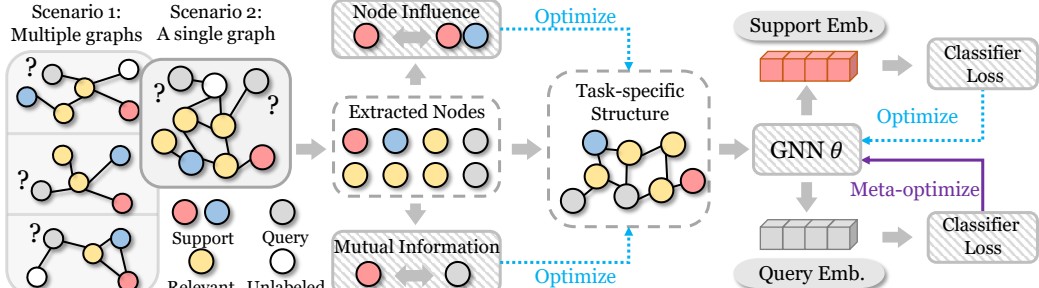

Figure 1: The overall framework of GLITTER. We first extract relevant nodes based on two strategies: local sampling and common sampling. Then we learn the task-specific structure with the extracted nodes along with support and query nodes based on node influence and mutual information. The learned structure will be used to generate node representations with a GNN. We further classify support nodes with a classifier, and the classification loss is used to optimize the GNN and the classifier. Finally, we meta-optimize the GNN and the classifier with the loss on query nodes.

## 3 Methodology

In this section, we introduce our framework that explores task-specific structures for different meta-tasks in graph few-shot learning. The detailed framework is illustrated in Figure 1. We first elaborate on the process of selecting relevant nodes based on node influence to construct the task-specific structure in each meta-task. Then we provide the detailed process of learning task-specific edge weights via maximizing node influence within classes and mutual information between query nodes and labels. Finally, we describe the meta-learning strategy used to optimize model parameters.

### 3.1 Selecting Nodes for Task-specific Structures

Given a meta-task $\mathcal{T} = \{\mathcal{S}, \mathcal{Q}\}$, we first aim to extract relevant nodes that are helpful for $\mathcal{T}$ and construct a task-specific structure $G_{\mathcal{T}}$ based on these nodes. In this way, we can reduce the impact of redundant information on the original graph and focus on meta-task $\mathcal{T}$. Nevertheless, it remains difficult to determine which nodes are useful for classification in $\mathcal{T}$. The reason is that the support nodes in $\mathcal{T}$ can be distributed across the original graph, which increases the difficulty of selecting nodes that are relevant to all these support nodes. Thus, we propose to leverage the concept of node influence to select relevant nodes. Here we first define node influence based on [18, 44, 34] as follows:

**Definition 1** (Node Influence). *The node influence from node $v_i$ to node $v_j$ is defined as $I(v_i, v_j) = \|\partial \mathbf{h}_i / \partial \mathbf{h}_j\|$, where $\mathbf{h}_i$ and $\mathbf{h}_j$ are the output representations of $v_i$ and $v_j$ in a GNN, respectively. $\partial \mathbf{h}_i / \partial \mathbf{h}_j$ is a Jacobian matrix, and the norm can be any specific subordinate norm.*

According to Definition 1, large node influence denotes that the representation of a node can be easily impacted by another node, thus rendering stronger correlations. Intuitively, we need to incorporate more nodes with large influence on the support nodes into $G_{\mathcal{T}}$. In this way, $G_{\mathcal{T}}$ can maintain the most crucial information that is useful for classification based on support nodes. To effectively select nodes with larger influence on the support nodes, we consider important factors that affect node influence. The following theorem provides a universal pattern for node influence on support nodes:

**Theorem 3.1.** *Consider the node influence from node $v_k$ to the $i$-th class (i.e., $C_i$) in a meta-task $\mathcal{T}$. Denote the geometric mean of the node influence values to all support nodes in $C_i$ as $I_{C_i}(v_k) = \sqrt[K]{\prod_{j=1}^{K} I(v_k, s_{i,j})}$, where $s_{i,j}$ is the $j$-th support node in $C_i$. Assume the node degrees are randomly distributed with the mean value as $\bar{d}$. Then, $\mathbb{E}(\log I_{C_i}(v_k)) \geq -\log \bar{d} \cdot \sum_{j=1}^{K} SPD(v_k, s_{i,j})/K$, where $SPD(v_k, s_{i,j})$ denotes the shortest path distance between $v_k$ and $s_{i,j}$.*

The proof is provided in Appendix A. Theorem 3.1 indicates that the lower bound of the node influence on a specific class is measured by its shortest path distances to all support nodes in this class. Therefore, to effectively select nodes with large influence on a specific class, we can choose nodes with small average shortest path distances to support nodes of this class. Based on this theorem, we propose two strategies, namely *local sampling* and *common sampling*, to select nodes for the

task-specific structure $G_{\mathcal{T}}$. In particular, we combine the selected nodes with support and query nodes in the meta-task (i.e., $\mathcal{S}$ and $\mathcal{Q}$) to obtain the final node set $\mathcal{V}_{\mathcal{T}} = \mathcal{V}_l \cup \mathcal{V}_c \cup \mathcal{S} \cup \mathcal{Q}$. Here $\mathcal{V}_l$ and $\mathcal{V}_c$ are the node sets extracted based on local sampling and common sampling, respectively. $\mathcal{S}$ and $\mathcal{Q}$ are the support set and the query set of $\mathcal{T}$, respectively. Then we introduce the two strategies in detail.

- **Local Sampling.** In this strategy, we extract the local neighbor nodes of support nodes within a specific distance (i.e., neighborhood size). Intuitively, the neighbor nodes can maintain a small shortest path distance to a specific support node. Therefore, by combining neighbor nodes of all support nodes in a class, we can obtain nodes with considerable node influence on this class without calculating the shortest path distances. Specifically, the extracted node set is denoted as $\mathcal{V}_l = \cup_{v_i \in \mathcal{S}} \mathcal{N}_l(v_i)$, where $\mathcal{N}_l(v_i) = \{u | d(u, v_i) \leq h\}$, and $h$ is the pre-defined neighborhood size.
- **Common Sampling.** In this strategy, we select nodes that maintain a small average distance to all nodes in the same class. In this way, the node influence on an entire class can be considered. Specifically, for each of $N$ classes in $\mathcal{T}$, we extract nodes with the smallest average distances to nodes in this support class. The overall extracted node set $\mathcal{V}_c$ can be presented as follows:

$$\mathcal{V}_c = \cup_{i=1}^N \underset{\mathcal{V}' \subset \mathcal{V}, |\mathcal{V}'|=C}{\mathrm{argmin}} \sum_{v \in \mathcal{V}'} \sum_{j=1}^K d(v, s_{i,j}), \tag{1}$$

where $s_{i,j}$ is the $j$-th node of the $i$-th class in $\mathcal{T}$. Here we extract $C$ nodes with the smallest sum of shortest path distances to nodes in each class. Then we aggregate these nodes into the final node set $\mathcal{V}_c$. In this way, we can select nodes with large influence on an entire class. As a result, the selected nodes will bear more crucial information for classifying a specific class. Note that since there are only $N$ classes in $\mathcal{T}$, the maximum size of $\mathcal{V}_c$ is $NC$, i.e., $|\mathcal{V}_c| \leq NC$.

**Edge Weight Functions.** With the extracted node set $\mathcal{V}_{\mathcal{T}}$, we intend to learn task-specific edge weights for $G_{\mathcal{T}}$. Intuitively, although the original structural information is crucial for classification, it can also be redundant for meta-task $\mathcal{T}$. Therefore, we propose to construct the edges based on both node representations and the shortest path distance between two nodes. In this way, the model will learn to maintain and learn beneficial edges for $\mathcal{T}$. Particularly, the edge weight starting from node $v_i$ to node $v_j$ is denoted as $a_{i,j} = (a_{i,j}^r + a_{i,j}^s)/2$, where $a_{i,j}^r$ and $a_{i,j}^s$ are learned by two functions that utilize node representations and structures as input, respectively.

- **Node representations as input.**

$$a_{i,j}^r = \exp\left(-\left\| \frac{\phi(\mathbf{W}_1 \mathbf{x}_i)}{\|\phi(\mathbf{W}_1 \mathbf{x}_i)\|_2} - \frac{\phi(\mathbf{W}_2 \mathbf{x}_j)}{\|\phi(\mathbf{W}_2 \mathbf{x}_j)\|_2} \right\|_2 \right), \tag{2}$$

where $\phi$ is a non-linear activation function, and $\mathbf{x}_i$ denotes the input feature vector of node $v_i$. $\mathbf{W}_1 \in \mathbb{R}^{d_a \times d}$ and $\mathbf{W}_2 \in \mathbb{R}^{d_a \times d}$ are learnable parameters, where $d_a$ is the dimension size of $\mathbf{W}_1 \mathbf{x}_i$ and $\mathbf{W}_2 \mathbf{x}_j$. $a_{i,j}^r$ is the edge weight between node $v_i$ and $v_j$ learned from node representations. Such a design naturally satisfies that $a_{i,j}^r \in (0, 1]$. Moreover, by introducing two weight matrices $\mathbf{W}_1$ and $\mathbf{W}_2$, the learned task-specific structured will be a directed graph to encode more information.

- **Structures as input.**

$$a_{i,j}^s = \mathrm{Sigmoid}\left(\psi\left(\mathrm{SPD}(v_i, v_j)\right)\right), \tag{3}$$

where $\psi$ is a learned function that outputs a scalar while utilizing the shortest path distance between $v_i$ and $v_j$ (i.e., $\mathrm{SPD}(v_i, v_j)$) on the original graph. In this way, we can preserve the structural information on the original graph by mapping the distance to a scalar. For example, if $\psi$ is learned as a decreasing function regarding the input $\mathrm{SPD}(v_i, v_j)$, the obtained task-specific structure will result in stronger correlations among nodes that are close to each other on the original graph.

### 3.2 Learning Task-specific Structures from Labeled Nodes

With the proposed functions for edge weights, we still need to optimize these weights to obtain the task-specific structure for $\mathcal{T}$. In particular, we can leverage the label information inside labeled nodes (i.e., support nodes in each meta-task). Intuitively, the task-specific structure should ensure that the learned representations of nodes in the same class are similar, so that the classification of this class will be easier. According to Definition 1, larger node influence represents stronger correlations between nodes, which will increase the similarity between the learned representations. Therefore,

we propose to enhance the node influence between support nodes in each class to optimize the task-specific structure. However, directly calculating the Jacobian matrix $\|\partial\mathbf{h}_i/\partial\mathbf{h}_j\|$ can result in an excessively large computational cost, especially when the representations are high-dimensional [34]. Hence, instead of directly computing the node influence, we propose to enhance the node influence based on the following theorem:

**Theorem 3.2** (Node Influence within Classes). *Consider the expectation of node influence from the $j$-th node $v_j$ to all nodes in its same class $\mathcal{C}$ on $G$, where $v_j \in \mathcal{C}$. Assume that $|\mathcal{C}| = K > 2$. The overall node influence is $\mathbb{E}\left(\sum_{v_i \in \mathcal{C}\backslash\{v_j\}} I(v_i, v_j)\right) = \sum_{v_i \in \mathcal{C}\backslash\{v_j\}} \frac{K-2}{n-1}\mathbf{h}_j^\top \cdot \mathbf{h}_i/\|\mathbf{h}_j\|^2 - \sum_{v_k \in \mathcal{V}\backslash\mathcal{C}} \frac{K-1}{n-1}\mathbf{h}_j^\top \cdot \mathbf{h}_k/\|\mathbf{h}_j\|^2$, where $n$ is the number of nodes in $G$.*

The proof is deferred to Appendix B. Theorem 3.2 states that maximizing the node influence within one class is equivalent to maximizing the similarity within classes while minimizing similarity between classes. Specifically, the similarity is measured by the dot product between representations of $v_j$ and other nodes. In this way, we can effectively optimize the edge weights by increasing the node influence within each class without a significant computational cost. The overall loss in meta-task $\mathcal{T}$ can be formulated as follows:

$$\mathcal{L}_N = -\frac{1}{N}\sum_{l=1}^{N}\sum_{v_j \in \mathcal{C}_l}\left(\sum_{v_i \in \mathcal{C}_l\backslash\{j\}}\frac{K-2}{|\mathcal{V}_\mathcal{T}|-1}\mathbf{h}_j^\top \cdot \mathbf{h}_i/\|\mathbf{h}_j\|^2 - \sum_{v_k \in \mathcal{V}\backslash\mathcal{C}}\frac{K-1}{|\mathcal{V}_\mathcal{T}|-1}\mathbf{h}_j^\top \cdot \mathbf{h}_k/\|\mathbf{h}_j\|^2\right),$$
(4)

where $\mathcal{C}_l$ denotes the node set of the $l$-th support class in meta-task $\mathcal{T}$, and $|\mathcal{C}_l| = K$.

## 3.3 Learning Task-specific Structures from Unlabeled Nodes

**Mutual Information.** Although we have leveraged the information in support nodes by enhancing node influence, the unlabeled nodes (i.e., query nodes) in $G_\mathcal{T}$ remain unused. Thus, we further propose to utilize the query nodes to improve the learning of task-specific structures. Specifically, we leverage the concept of mutual information (MI) [12] by maximizing the MI between query nodes and their potential labels. Formally, the MI can be represented as follows:

$$\mathcal{L}_M := -\mathcal{I}(X_Q; Y_Q) = -\mathcal{H}(Y_Q) + \mathcal{H}(Y_Q|X_Q)$$
$$= \sum_{j=1}^{N}\bar{p}_j \log \bar{p}_j - \frac{1}{|\mathcal{Q}|}\sum_{i\in\mathcal{Q}}\sum_{j=1}^{N}p_{ij}\log p_{ij},$$
(5)

where $p_{ij} := P(y_i = j|\mathbf{x}_i)$ and $\bar{p}_j = \sum_{i\in\mathcal{Q}}p_{ij}/|\mathcal{Q}|$. $\mathcal{Q}$ is the query node set in meta-task $\mathcal{T}$. Here $X_Q$ and $Y_Q$ are the features and labels of query nodes, respectively. In particular, by maximizing the mutual information between query nodes and their labels, the learned task-specific structure can effectively leverage the information from the query set.

**Transformed Markov Chain.** Although the strategy of maximizing MI between $X_Q$ and $Y_Q$ has proven to be effective in few-shot learning, it remains non-trivial to utilize such a strategy for learning task-specific structures. Specifically, the crucial part is to model the dependency of classification probabilities (i.e., $p_{ij}$) on the learned task-specific structure. In this way, we can leverage the mutual information to help optimize the task-specific structure. To estimate the classification probability, we propose to utilize node influence inferred from the task-specific structure in each meta-task. Intuitively, we can assume that the higher node influence between two nodes can represent a larger probability that they share the same class. As a result, by estimating the node influence between a query node and a support node, we can optimize the task-specific structure based on Eq. (5). However, it still remains challenging to obtain an exact value of the node influence due to the potential computation cost of calculating the Jacobian matrix. Therefore, we propose to estimate node influence via the absorbing probability in a Markov chain [31]. Particularly, we create a Markov chain with absorbing states, where each state corresponds to a node on $G_\mathcal{T}$. Here the support nodes in $G_\mathcal{T}$ will be transformed into the absorbing states. Denoting the current adjacency matrix for $G_\mathcal{T}$ as $\mathbf{A}$, we obtain the transition matrix of the Markov chain from $\mathbf{A}$ as follows: (1) The transition probability from a support node to other nodes is 0 (i.e., the absorbing state). (2) Each entry is row-wise normalized to ensure that the sum of each row equals 1 (i.e., the transition probabilities starting from one node should sum up to 1). Denoting the obtained transition probability matrix as $\tilde{\mathbf{A}}$.

Here we theoretically validate that the node influence between a query node and a support node can be effectively and efficiently estimated by the absorbing probability.

**Theorem 3.3** (Absorbing Probabilities and Node Influence). *Consider the Markov chain with a transition matrix $\tilde{\mathbf{A}}$ derived from a graph $G$. Denote the probability of being absorbed in the $j$-th state (absorbing state) when starting from the $i$-th state as $b_{i,j}$. Then, $I(v_i, v_j) = b_{i,j}$, where $I(v_i, v_j)$ is the node influence from the $i$-th node $v_i$ to the $j$-th node $v_j$ on $G$.*

The proof is provided in Appendix C. Theorem 3.3 provides that with the learned task-specific structure in meta-task $\mathcal{T}$, the node influence of a support node on a query node equals the absorbing probability starting from the query node to the support node in the Markov chain. In consequence, we can leverage the property of Markov chains to provide an efficient approximation for node influence.

**Theorem 3.4** (Approximation of Absorbing Probabilities). *Denote $t$ as the number of non-absorbing states in the Markov chain, i.e., $t = |\mathcal{V}_{\mathcal{T}}| - |\mathcal{S}|$, where $|\mathcal{V}_{\mathcal{T}}|$ is the node set in $G_{\mathcal{T}}$. Denote $\mathbf{Q} \in \mathbb{R}^{t \times t}$ as the transition probability matrix for non-absorbing states in the Markov chain. Estimating $b_{i,j}$ with $\tilde{b}_{i,j} = \sum_{k=1}^{t} \tilde{\mathbf{A}}_{k,j} \cdot \sum_{h=0}^{m} \left(\mathbf{Q}^h\right)_{i,k}$, then the absolute error is upper bounded as $|b_{i,j} - \tilde{b}_{i,j}| \leq C/t^{m-1}(t-1)$, where $m$ controls the upper bound, and $C$ is a constant.*

The proof is provided in Appendix D. Theorem 3.4 indicates that the estimation error of node influence is upper bounded by $C/t^{m-1}(t-1)$, which is controlled by $m$. Therefore, we can adjust the value of $m$ to fit in different application scenarios. With Theorem 3.3 and Theorem 3.4, we can estimate the value of the classification probability as follows:

$$P(y_i = j|\mathbf{x}_i) = \sum_{k=1}^{t} \tilde{\mathbf{A}}_{k,j} \cdot \sum_{h=0}^{m} (\mathbf{Q}^h)_{i,k} \tag{6}$$

As a result, we can optimize the objective in Eq. (5) according to the estimated node influence. In this way, we can adjust edge weights to reach a state that is tailored for this meta-task $\mathcal{T}$ and beneficial for the following classification of query nodes.

## 3.4 Meta Optimization

To effectively accumulate learned meta-knowledge across a variety of meta-tasks, we adopt the prevalent meta-optimization strategy based on MAML [14]. Generally, MAML aims at learning the unique update scheme based on each meta-task for fast adaptation to novel meta-tasks. Specifically, for each meta-task, we first perform several update steps to learn task-specific structures based on the two structure losses: $\theta_S^{(i)} = \theta_S^{(i-1)} - \alpha \nabla \mathcal{L}_S^{(i)}$, where $\theta_S$ denotes the parameters used to learn task-specific structures, and $\mathcal{L}_S^{(i)} = \mathcal{L}_N^{(i)} + \mathcal{L}_M^{(i)}$. During each update step, we leverage a GNN to learn node representations from the current task-specific structure, followed by an MLP layer to classify the support nodes: $\theta_G^{(i)} = \theta_G^{(i-1)} - \alpha \nabla \mathcal{L}_{support}^{(i)}$, where $\mathcal{L}_{support}^{(i)} = -\sum_{k \in \mathcal{S}} \sum_j y_{k,j} \log p_{k,j}^{(i)}$ is the cross-entropy loss, and $\alpha$ is the base learning rate. Here $y_{k,j} \in \{0,1\}$ and $y_{k,j} = 1$ if the $k$-th node in $\mathcal{S}$ (i.e., the support set) belongs to the $j$-th support class in meta-task $\mathcal{T}$; otherwise $y_{k,j} = 0$. $p_{k,j}^{(i)}$ is the $j$-the entry of the class distribution vector $\mathbf{p}_k^{(i)}$ calculated by $\mathbf{p}_k^{(i)} = \text{Softmax}(\text{MLP}(\mathbf{h}_k^{(i)}))$, where $\mathbf{h}_k^{(i)}$ is extracted from the learned node representations $\mathbf{H}^{(i)} = \text{GNN}(\mathbf{A}^{(i)}, \mathbf{X})$. Here $\mathbf{A}^{(i)}$ is the edge weights during the $i$-th step. $\theta_G$ denotes the parameters of the GNN and the MLP classifier. After repeating these steps for $\eta$ times, the loss on the query set will be used for the meta-update: $\theta_G = \theta_G^{(\eta)} - \beta_1 \nabla \mathcal{L}_{query}^{(\eta)}$ and $\theta_S = \theta_S^{(\eta)} - \beta_2 \nabla \mathcal{L}_S^{(\eta)}$, where $\beta_1$ and $\beta_2$ are the meta-learning rates, and $\mathcal{L}_{query}^{(\eta)}$ is the cross-entropy loss calculated on query nodes. For the meta-test tasks, we will use the final meta-updated parameters to evaluate the model.

## 4 Experiments

**Experimental settings.** The training process of our framework is conducted on a series of meta-training tasks. After training, the model will be evaluated on a specific number of meta-test tasks. Here we introduce four few-shot node classification settings for the experiments. (1) Shared graphs with disjoint labels. Under this setting, the meta-training and meta-test tasks are sampled from the same graph set. The labels in meta-test tasks are disjoint from those in meta-training

tasks. (2) Disjoint graphs with shared labels. Under this setting, the meta-training tasks and meta-test tasks are from disjoint graph sets. In other words, the graphs in meta-test tasks will be new graphs unseen during training. The labels are shared during training and test, which slightly decreases the difficulty. (3) Disjoint graphs with disjoint labels. Under this setting, both the graph set and the label set of meta-test tasks are disjoint from those in meta-training tasks. That being said, the model is required to handle new labels on graphs unseen during training, which renders the highest difficulty. (4) Single graph with disjoint labels. This setting is widely adopted in existing graph few-shot learning methods [10, 36, 48]. Under this setting, all nodes will be sampled from the same graph. Note that setting (1) and setting (2) are proposed in G-Meta [18], while setting (3) is a new setting introduced in this paper. Detailed experimental and parameter settings are provided in Appendix E.

**Datasets.** To evaluate the performance of GLITTER on the few-shot node classification task, we conduct experiments on five prevalent real-world graph datasets, two for the multiple-graph setting and three for the single-graph setting. The detailed statistics are provided in Table 1. For the multiple-graph setting, we use the following two datasets: (1) Tissue-PPI [49] consists of 24 protein-protein interaction (PPI) networks obtained from different types of tissues. Specifically, node features are gene signatures, and labels are gene ontology functions. This dataset consists of ten binary classification tasks. (2) Fold-PPI [18] contains of 144 tissue PPI networks, where the node labels are protein structures in SCOP database [1]. Moreover, the node features are conjoint triad protein descriptors [26]. There are 29 classes in this dataset. For the single-graph setting, we leverage three datasets: (1) DBLP [30] is a citation network with nodes and edges representing papers and citation relations, respectively. The node features are based on the paper abstracts, and the labels are determined by the paper venues. This dataset consists of 137 classes. (2) Cora-full [2] is also a citation network which contains 70 classes. (3) ogbn-arxiv [17] is a citation network that consists of 20 classes and significantly more nodes than other datasets.

Table 1: Statistics of five node classification datasets.

| Dataset | # Graphs | # Nodes | # Edges | # Features | # Class |
|---------|----------|---------|---------|------------|---------|
| Tissue-PPI | 24 | 51,194 | 1,350,412 | 50 | 10 |
| Fold-PPI | 144 | 274,606 | 3,666,563 | 512 | 29 |
| DBLP | 1 | 40,672 | 288,270 | 7,202 | 137 |
| Cora-full | 1 | 19,793 | 65,311 | 8,710 | 70 |
| ogbn-arxiv | 1 | 169,343 | 1,166,243 | 128 | 40 |

**Baseline methods.** (1) KNN [13] trains a GNN based on all samples in training graphs to learn node representations. Then it classifies query samples according to the labels in the support set of each meta-task. (2) ProtoNet [27] learns the prototype of each class by averaging its node representations. Then it classifies query samples based on their distances to the prototypes. (3) MAML [14] adopts the meta-learning strategy and learns node representations via GNNs. (4) Meta-GNN combines MAML with Simple Graph Convolution (SGC) [40]. (5) G-Meta [18] learns node representations from extracted local subgraphs and further adopts MAML for meta-learning. (6) GPN [10] learns prototypes based on node importance and classifies query nodes according to their distances to the learned prototypes. (7) RALE [21] learns node dependencies based on node locations on the graph. It is noteworthy that Meta-GNN, GPN, and RALE only work for the single-graph setting, while other baseline methods can be applied to both single-graph and multiple-graph settings.

## 4.1 Main Results

Table 2 and Table 3 present the performance comparison of our framework GLITTER and all baselines on few-shot node classification under the multiple-graph setting and the single-graph setting, respectively. Specifically, for the multiple-graph task, we conduct experiments under three settings (i.e., shared graphs with disjoint labels, disjoint graphs with shared labels, and disjoint graphs with disjoint labels). For the single-graph task, we choose two different few-shot settings to obtain a more comprehensive comparison: 5-way 3-shot (i.e., $N = 5$ and $K = 3$) and 10-way 3-shot (i.e., $N = 10$ and $K = 3$). For the evaluation metric, we utilize the average classification accuracy over 10 repetitions. From Table 2 and Table 3, we can obtain the follow observations: (1) Our framework GLITTER outperforms all other baselines in all multiple-graph and single-graph datasets. GLITTER also consistently achieves the best results under different settings, which validates the superiority of GLITTER on few-shot node classification problems. (2) When the multiple-graph

setting is conducted with disjoint graphs, GLITTER has the least performance drop compared with other baselines. This is because the learned task-specific structure is tailored for each meta-task and is thus more capable in the multiple-graph setting, where the structures in meta-tasks across graphs can vary greatly. (3) The performance improvement of GLITTER over other baselines is relatively larger on Fold-PPI and DBLP. This is because these datasets consist of more classes than other datasets under the same setting. Since the classes in different meta-tasks are more diversely distributed, the task-specific structures learned by GLITTER will provide better performance in this situation. (4) When the value of $N$ increases (i.e., larger class set in each meta-task), the performance of all baselines drops significantly due to the higher classification difficulty. Nevertheless, GLITTER consistently outperforms the other baselines. The reason is that a larger class set in a meta-task will require more specific representations for classification due to the variety of classes.

Table 2: The few-shot node classification results (accuracy in %) under the multiple-graph setting.

| Dataset | Tissue-PPI | | | Fold-PPI | | |
|---|---|---|---|---|---|---|
| Graph Setting | Shared | Disjoint | Disjoint | Shared | Disjoint | Disjoint |
| Label Setting | Disjoint | Shared | Disjoint | Disjoint | Shared | Disjoint |
| KNN [13] | 62.3±3.1 | 63.6±2.3 | 56.6±3.2 | 40.0±3.4 | 50.2±2.2 | 38.6±2.8 |
| ProtoNet [27] | 59.0±3.3 | 61.8±2.7 | 55.7±3.2 | 37.4±2.0 | 46.5±2.0 | 34.7±3.1 |
| MAML [14] | 63.8±4.3 | 68.4±3.7 | 56.0±4.0 | 41.0±3.0 | 49.3±2.8 | 42.5±4.4 |
| G-Meta [18] | 64.6±4.9 | 69.9±4.1 | 58.1±2.4 | 46.1±5.0 | 54.7±4.3 | 50.3±5.2 |
| GLITTER | **69.7**±3.9 | **73.1**±4.8 | **60.3**±3.1 | **53.3**±3.6 | **61.3**±2.9 | **54.7**±4.0 |

Table 3: The few-shot node classification results (accuracy in %) under the single-graph setting.

| Dataset | DBLP | | Cora-full | | ogbn-arxiv | |
|---|---|---|---|---|---|---|
| Setting | 5-way 3-shot | 10-way 3-shot | 5-way 3-shot | 10-way 3-shot | 5-way 3-shot | 10-way 3-shot |
| KNN [13] | 63.4±3.3 | 54.1±2.5 | 56.9±2.1 | 42.4±2.7 | 45.4±3.3 | 35.7±2.4 |
| ProtoNet [27] | 58.5±2.6 | 51.3±3.3 | 49.6±2.2 | 41.4±1.9 | 42.9±3.0 | 34.6±3.0 |
| MAML [14] | 60.4±4.5 | 52.3±3.5 | 54.0±4.4 | 38.1±3.7 | 43.0±3.1 | 34.1±3.4 |
| Meta-GNN [48] | 66.4±2.8 | 58.0±2.6 | 59.7±2.7 | 42.5±3.6 | 48.8±2.4 | 35.0±4.8 |
| G-Meta [18] | 73.1±3.4 | 57.5±5.0 | 63.7±4.0 | 47.6±4.9 | 50.8±4.8 | 38.2±4.9 |
| GPN [10] | 75.4±2.3 | 64.0±3.8 | 62.1±3.4 | 47.3±4.4 | 55.3±3.1 | 36.3±4.6 |
| RALE [21] | 74.9±3.2 | 65.1±5.7 | 63.4±4.0 | 45.2±3.6 | 54.7±3.9 | 38.4±5.1 |
| GLITTER | **79.3**±3.1 | **69.5**±5.2 | **66.2**±4.4 | **52.0**±2.6 | **58.5**±4.7 | **44.2**±3.8 |

## 4.2 Ablation Study

In this section, we perform a series of ablations studies to evaluate the effectiveness of different components in our framework GLITTER. The results are presented in Figure 2. Specifically, we compare our proposed framework GLITTER with three degenerate versions: (1) GLITTER without sampling relevant nodes (i.e., the task-specific structure only consists of support nodes and query nodes), denoted as GLITTER\S; (2) GLITTER without enhancing the node influence within classes, denoted as GLITTER\N; (3) GLITTER without maximizing the mutual information between query nodes and their potential labels, denoted as GLITTER\M. From Figure 2, we can obtain several meaningful observations. First, our framework consistently outperforms all variants with certain components removed, which demonstrates that each module in GLITTER plays a crucial role in learning task-specific structures. Second, removing the relevant nodes deteriorates the performance under the multiple-graph setting with disjoint graphs more than that on shared graphs. The reason is meta-knowledge from training graphs can be uninformative for a novel graph during evaluation. As a result, the relevant nodes on the novel graph become more crucial for learning task-specific structures. Third, the performance decreases differently by removing the node influence module or the mutual information module. Specifically, removing node influence generally causes a more significant drop when $N$ increases. This is due to that a larger class set in each meta-task requires more distinct node representations in different classes. Therefore, enhancing the node influence can provide better performance by enlarging correlations within classes.

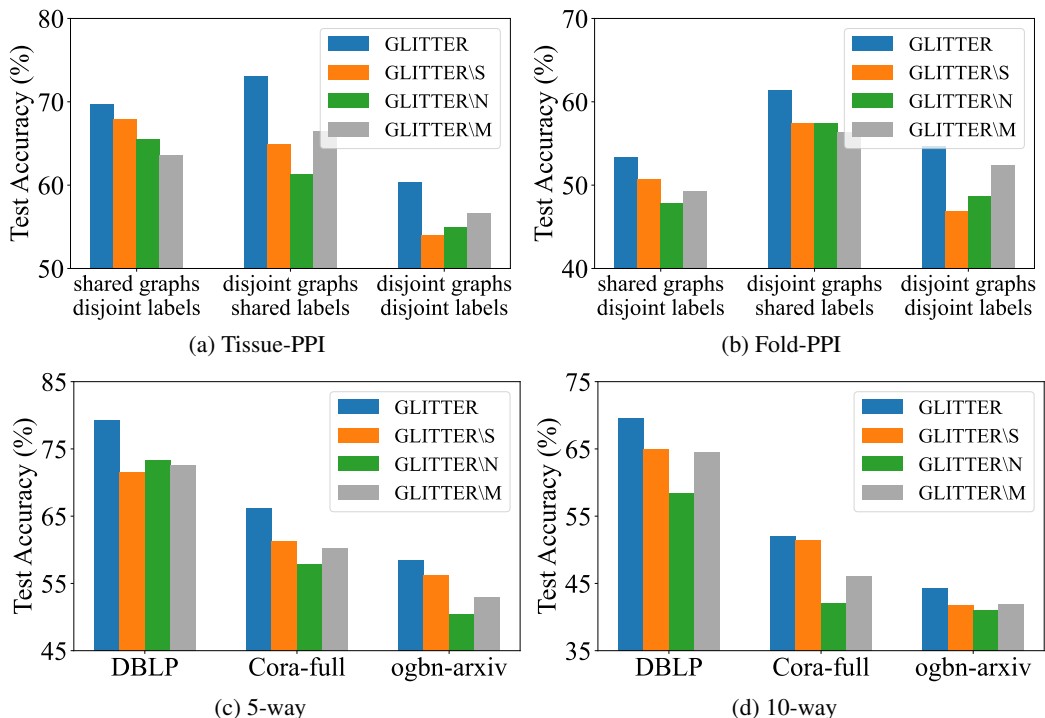

Figure 2: The ablation study of GLITTER on both single-graph and multiple-graph settings.

## 5 Related Work

**Few-shot Learning on Graphs.** Few-shot learning aims at effectively solving tasks with scarce labeled samples as references. The general approach is to learn meta-knowledge from tasks with abundant labeled samples and generalize it to a new task with few labeled samples. Few-shot learning frameworks generally adopt the meta-learning strategy, which means both training and test processes will be conducted on a series of learning tasks. Existing few-shot learning methods can be broadly divided into two categories: (1) *Metric-based* approaches, which aim at learning a metric function between the query set and the support set for classification [33, 20, 28, 27]. (2) *Optimization-based* approaches, which focus on updating model parameters according to gradients calculated on the support set in each meta-task [22, 25, 14, 23]. Recently, many works propose to incorporate existing few-shot learning frameworks into graph few-shot learning problems [21, 36, 9, 42, 7, 39, 29, 38]. For example, GPN [10] leverages the importance of nodes and combines Prototypical Networks [27] to solve few-shot node classification problem. G-Meta [18] extracts local subgraphs for nodes for fast adaptations across multiple graphs. TENT [37] proposes to mitigate the adverse impact of task variance with proposed node-level and class-level adaptations.

**Graph Neural Networks.** Recently, many researchers have focused on using Graph Neural Networks (GNNs) to learn effective representation for nodes on graphs [5, 44, 41, 4, 46, 8, 41, 11]. With GNNs, models will gradually learn comprehensive node representations based on the specific downstream tasks [16]. In particular, GNNs learn node representations from neighbor nodes via an information aggregation mechanism. As a result, GNNs can incorporate both structural information and node feature information into the representation learning process. For example, Graph Convolutional Networks (GCNs) [19] leverage a first-order approximation to learn node representations with graph convolution layers in a simplified manner. Graph Attention Networks (GATs) [32] aim to learn adjustable weights for neighboring nodes via the attention mechanism to aggregate features of neighboring nodes in an adaptive manner.

## 6 Conclusion

In this paper, we propose to learn task-specific structures for each meta-task in graph few-shot learning, so that the learned node representations will be tailored for each meta-task. To handle the

variety of nodes among meta-tasks, we propose a novel framework, GLITTER, to extract important nodes and learn task-specific structures based on node influence and mutual information. As a result, our framework can adaptively learn node representations to promote classification performance in each meta-task. We conduct extensive experiments on five node classification datasets under both the single-graph and multiple-graph settings, and the results validate the superiority of our framework over other state-of-the-art baselines.

## Acknowledgement

This work is supported by the National Science Foundation under grants IIS-2006844, IIS-2144209, IIS-2223769, CNS-2154962, and BCS-2228534, the JP Morgan Chase Faculty Research Award, the Cisco Faculty Research Award, and Jefferson Lab subcontract JSA-22-D0311.

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
