## A Theorem 3.1 and Proof

Before proving Theorem 3.1, we first provide a lemma that demonstrate the lower bound of node influence between two nodes. In the following proof, we will follow [18] and [44] to use GCNs [19] as the exemplar GNN for simplicity. Moreover, it is noteworthy that our proof can be naturally generalized to different types of GNNs (e.g., GAT [32] and GraphSAGE [16]) by assigning different values for edge weights. Specifically, the $l$-th layer propagation process can be represented as $\mathbf{H}^{(l+1)} = \sigma(\hat{\mathbf{A}}\mathbf{H}^{(l)}\mathbf{W}^{(l)})$. Here $\mathbf{H}^{(l)}$ and $\mathbf{W}^{(l)}$ denote the node representation and weight parameter matrices, respectively. $\hat{\mathbf{A}} = \mathbf{D}^{-1}\mathbf{A}$ is the adjacency matrix after row normalization, which means

each row of $\hat{\mathbf{A}}$ sums up to 1. Following [18], [34], and [44], we set $\sigma$ as an identity function and $\mathbf{W}$ an identity matrix. Moreover, we assume that the propagation process is conducted in a sufficient number of steps. As a result, the output representation of one node can be represented by representations of its neighbor nodes.

**Lemma A.1.** *Consider the expectation of node influence between node $v_i$ and node $v_j$, i.e., $\mathbb{E}\left(I(v_i, v_j)\right)$. Assume that the node degrees are distributed uniformly for each node with the mean value $\bar{d}$. Then, $\mathbb{E}\left(I(v_i, v_j)\right) \geq \bar{d}^{(SPD(v_i, v_j))}$, where $SPD(v_i, v_j)$ is the shortest path distance between $v_i$ and $v_j$.*

*Proof.* Based on the GCN propagation strategy, we know that on $G$, the representation of node $v_i$ can be represented as

$$\mathbf{h}_i = \frac{1}{D_{ii}} \sum_{k \in \mathcal{N}(i)} a_{ik} \mathbf{h}_k,$$

where $\mathcal{N}(i)$ denotes the set of neighbor nodes of node $v_i$. Then we can expand the equation by incorporating 2-hop neighbors of node $v_i$:

$$\mathbf{h}_i = \frac{1}{D_{ii}} \sum_{k \in \mathcal{N}(i)} a_{ik} \frac{1}{D_{kk}} \sum_{l \in \mathcal{N}(k)} a_{kl} \mathbf{h}_l.$$

In this way, the expectation of node influence $I_{i,j} = \|\partial \mathbf{h}_i / \partial \mathbf{h}_j\|$ can be represented as:

$$
\begin{aligned}
\left\|\frac{\partial \mathbf{h}_i}{\partial \mathbf{h}_j}\right\| &= \left\| \frac{\partial}{\partial \mathbf{h}_j} \left( \frac{1}{D_{ii}} \sum_{k \in \mathcal{N}(i)} a_{ik} \frac{1}{D_{kk}} \sum_{l \in \mathcal{N}(k)} a_{kl} \cdots \frac{1}{D_{mm}} \sum_{o \in \mathcal{N}(m)} a_{mo} \mathbf{h}_o \right) \right\| \\
&= \left\| \frac{\partial}{\partial \mathbf{h}_j} \left( \left( \frac{1}{D_{ii}} a_{ik_1^1} \frac{1}{D_{k_1^1 k_1^1}} a_{k_1^1 k_2^1} \cdots \frac{1}{D_{k_{n_1}^1 k_{n_1}^1}} a_{k_{n_1}^1 j} \mathbf{h}_j \right) \right. \right. \\
&\quad \left. \left. + \cdots + \left( \frac{1}{D_{ii}} a_{ik_1^m} \frac{1}{D_{k_1^m k_1^m}} a_{k_1^m k_2^m} \cdots \frac{1}{D_{k_{n_m}^m k_{n_m}^m}} a_{k_{n_m}^m j} \mathbf{h}_j \right) \right) \right\|.
\end{aligned}
\tag{7}
$$

In the above equation, we first substitute the term $\mathbf{h}_i$ by the iterative expansion of neighbors. In this expansion, we only keep $m$ paths that start from $v_i$ to $v_j$, where $n_i$ is the number of intermediate nodes on the $i$-th path. The reason is that since we are considering the gradient between $v_i$ and $v_j$, the derivative on paths that do not contain $v_j$ will be 0 and thus can be ignored. Then we can further extract the common term $\|\partial \mathbf{h}_j / \partial \mathbf{h}_j\|$:

$$
\begin{aligned}
\left\|\frac{\partial \mathbf{h}_i}{\partial \mathbf{h}_j}\right\| &= \left\|\frac{\partial \mathbf{h}_j}{\partial \mathbf{h}_j}\right\| \cdot \left( \left( \frac{1}{D_{ii}} a_{ik_1^1} \frac{1}{D_{k_1^1 k_1^1}} a_{k_1^1 k_2^1} \cdots \frac{1}{D_{k_{n_1}^1 k_{n_1}^1}} a_{k_{n_1}^1 j} \right) \right. \\
&\quad \left. + \cdots + \left( \frac{1}{D_{ii}} a_{ik_1^m} \frac{1}{D_{k_1^m k_1^m}} a_{k_1^m k_2^m} \cdots \frac{1}{D_{k_{n_m}^m k_{n_m}^m}} a_{k_{n_m}^m j} \right) \right) \\
&= \left( \frac{1}{D_{ii}} a_{ik_1^1} \frac{1}{D_{k_1^1 k_1^1}} a_{k_1^1 k_2^1} \cdots \frac{1}{D_{k_{n_1}^1 k_{n_1}^1}} a_{k_{n_1}^1 j} \right) \\
&\quad + \cdots + \left( \frac{1}{D_{ii}} a_{ik_1^m} \frac{1}{D_{k_1^m k_1^m}} a_{k_1^m k_2^m} \cdots \frac{1}{D_{k_{n_m}^m k_{n_m}^m}} a_{k_{n_m}^m j} \right).
\end{aligned}
\tag{8}
$$

In this equation, we first utilize $\|\partial \mathbf{h}_j / \partial \mathbf{h}_j\| = 1$. This is because $\|\partial \mathbf{h}_j / \partial \mathbf{h}_j\| = \|\mathbf{I}\| = \sup_{\|\mathbf{h}\|=1}\{\|\mathbf{I}\mathbf{h}\|\} = 1$. The resulting term is the expectation that sums up the node degree products of all paths between $v_i$ and $v_j$. Therefore, it is larger than the value on the path with the maximum

node degree product:

$$\left\|\frac{\partial \mathbf{h}_i}{\partial \mathbf{h}_j}\right\| \geq \max\left(\left(\frac{1}{D_{ii}}a_{ik_1^1}\frac{1}{D_{k_1^1 k_1^1}}a_{k_1^1 k_2^1}\cdots\frac{1}{D_{k_{n_1}^1 k_{n_1}^1}}a_{k_{n_1}^1 j}\right)\right.$$
$$\left.,\cdots,\left(\frac{1}{D_{ii}}a_{ik_1^m}\frac{1}{D_{k_1^m k_1^m}}a_{k_1^m k_2^m}\cdots\frac{1}{D_{k_{n_m}^m k_{n_m}^m}}a_{k_{n_m}^m j}\right)\right). \tag{9}$$
$$=\frac{1}{D_{ii}}a_{ik_1^*}\frac{1}{D_{k_1^* k_1^*}}a_{k_1^* k_2^*}\cdots\frac{1}{D_{k_{n_*}^* k_{n_*}^*}}a_{k_{n_*}^* j}.$$

If we assume the node degrees are uniformly distributed, then the expectation of node degree products on this path $p_*$ is $\bar{d}^{(n_*+1)}$, where $n_* + 1$ is the length, and $\bar{d}$ is the expectation of node degrees. Moreover, we know $p_*$ is exactly the shortest path between $v_i$ and $v_j$. Therefore,

$$\mathbb{E}\left(\left\|\frac{\partial \mathbf{h}_i}{\partial \mathbf{h}_j}\right\|\right) \geq \left(1/\bar{d}\right)^{(n_*+1)} = \bar{d}^{-(\mathrm{SPD}(v_i,v_j))}, \tag{10}$$

where $\mathrm{SPD}(v_i, v_j)$ denotes the shortest path distance between node $v_i$ and node $v_j$. $\qquad\square$

Now with Lemma A.1, we can prove Theorem 3.1.

**Theorem 3.1.** *Consider the node influence from node $v_k$ to the $i$-th class (i.e., $C_i$) in a meta-task $\mathcal{T}$. Denote the geometric mean of the node influence values to all support nodes in $C_i$ as $I_{C_i}(v_k) = \sqrt[K]{\prod_{j=1}^{K} I(v_k, s_{i,j})}$, where $s_{i,j}$ is the $j$-th support node in $C_i$. Assume the node degrees are randomly distributed with the mean value as $\bar{d}$. Then, $\mathbb{E}(\log I_{C_i}(v_k)) \geq -\log \bar{d}\cdot\sum_{j=1}^{K} SPD(v_k, s_{i,j})/K$, where $SPD(v_k, s_{i,j})$ denotes the shortest path distance between $v_k$ and $s_{i,j}$.*

*Proof.* We know $\log I_C(v_k)$ can be represented as follows:

$$\log I_C(v_k) = \frac{1}{K}\sum_{j=1}^{K}\log I(v_k, s_{i,j}) \tag{11}$$

Based on Lemma A.1, we know:

$$\mathbb{E}\left(\log I_C(v_k)\right) = \frac{1}{K}\sum_{j=1}^{K}\log \mathbb{E}\left(I(v_k, s_{i,j})\right) \geq \frac{1}{K}\sum_{j=1}^{K}-\mathrm{SPD}(v_k, s_{i,j})\cdot\log\bar{d}, \tag{12}$$

where $\mathrm{SPD}(v_i, v_j)$ denotes the shortest path distance between node $v_i$ and node $v_j$. By rearranging the term, we can obtain the final inequality:

$$\mathbb{E}\left(\log I_C(v_k)\right) \geq -\frac{\log\bar{d}}{K}\sum_{j=1}^{K}\mathrm{SPD}(v_k, s_{i,j}). \tag{13}$$

$\qquad\square$

## B  Theorem 3.2 and Proof

**Theorem 3.2** (Node Influence within Classes). *Consider the expectation of node influence from the $j$-th node $v_j$ to all nodes in its same class $\mathcal{C}$ on $G$, where $v_j \in \mathcal{C}$. Assume that $|\mathcal{C}| = K > 2$. The overall node influence is $\mathbb{E}\left(\sum_{v_i\in\mathcal{C}\backslash\{v_j\}} I(v_i, v_j)\right) = \sum_{v_i\in\mathcal{C}\backslash\{v_j\}}\frac{K-2}{n-1}\mathbf{h}_j^\top\cdot\mathbf{h}_i/\|\mathbf{h}_j\|^2 - \sum_{v_k\in\mathcal{V}\backslash\mathcal{C}}\frac{K-1}{n-1}\mathbf{h}_j^\top\cdot\mathbf{h}_k/\|\mathbf{h}_j\|^2$, where $n$ is the number of nodes in $G$.*

*Proof.* Following the expansion of node representations based on the neighbors in Lemma A.1, we can represent $\mathbf{h}_i$ as follows:

$$\mathbf{h}_i = \sum_{k=1}^{n} I_{ik}\mathbf{h}_k, \tag{14}$$

where $I_{ik}$ is node influence from $v_i$ to $v_k$. Then, we have

$$\mathbf{h}_j^\top \cdot \mathbf{h}_i/\|\mathbf{h}_j\|^2 = \sum_{k=1}^n I_{ik}\mathbf{h}_j^\top \cdot \mathbf{h}_k/\|\mathbf{h}_j\|^2$$

$$= I_{ij}\mathbf{h}_j^\top \cdot \mathbf{h}_j/\|\mathbf{h}_j\| + \sum_{k=1,k\neq j}^n I_{ik}\mathbf{h}_j^\top \cdot \mathbf{h}_k/\|\mathbf{h}_j\|^2 \tag{15}$$

$$= I_{ij} + \sum_{k=1,k\neq j}^n I_{ik}\mathbf{h}_j^\top \cdot \mathbf{h}_k/\|\mathbf{h}_j\|^2.$$

We can further sum up the node influence from nodes in the same class of $v_j$ to it since we aim to calculate the total node influence of a set of support nodes. Here we denote the support nodes set of $v_j$ class as $\mathcal{C}$, where $|\mathcal{C}| = K$. Therefore, we can obtain:

$$\sum_{i\in\mathcal{C}\setminus\{j\}} \mathbf{h}_j^\top \cdot \mathbf{h}_i/\|\mathbf{h}_j\|^2 = \sum_{i\in\mathcal{C}\setminus\{j\}} I_{ij} + \sum_{i\in\mathcal{C}\setminus\{j\}}\sum_{k\in\mathcal{V}\setminus(\{i\}\cup\{j\})} I_{ik}\mathbf{h}_j^\top \cdot \mathbf{h}_k/\|\mathbf{h}_j\|^2 \tag{16}$$

We can move the same terms from the RHS to the LHS:

$$\sum_{i\in\mathcal{C}\setminus\{j\}} \mathbf{h}_j^\top \cdot \mathbf{h}_i/\|\mathbf{h}_j\|^2 - \sum_{i\in\mathcal{C}\setminus\{j\}}\sum_{k\in\mathcal{C}\setminus\{i\}} I_{ik}\mathbf{h}_j^\top \cdot \mathbf{h}_k/\|\mathbf{h}_j\|^2$$

$$= \sum_{i\in\mathcal{C}\setminus\{j\}} I_{ij} + \sum_{i\in\mathcal{C}\setminus\{j\}}\sum_{k\in\mathcal{V}\setminus\mathcal{C}} I_{ik}\mathbf{h}_j^\top \cdot \mathbf{h}_k/\|\mathbf{h}_j\|^2 \tag{17}$$

Rearranging the LHS, we can obtain:

$$LHS = \sum_{i\in\mathcal{C}\setminus\{j\}} \left(1 - \sum_{k\in\mathcal{C}\setminus(\{i\}\cup\{j\})} a_{ki}\right) \mathbf{h}_j^\top \cdot \mathbf{h}_i/\|\mathbf{h}_j\|^2 \tag{18}$$

Rearranging the RHS, we can obtain:

$$RHS = \sum_{i\in\mathcal{C}\setminus\{j\}} I_{ij} + \sum_{k\in\mathcal{V}\setminus\mathcal{C}}\sum_{i\in\mathcal{C}\setminus\{j\}} I_{ik}\mathbf{h}_j^\top \cdot \mathbf{h}_k/\|\mathbf{h}_j\|^2 \tag{19}$$

We further assume that the expectation of each $I_{ij}$ is $1/(n-1)$ since the sum of probabilities of all paths starting from $v_i$ equals 1. Therefore,

$$\mathbb{E}\left(\sum_{i\in\mathcal{C}\setminus\{j\}} I_{ij}\right) = \mathbb{E}\left(\sum_{i\in\mathcal{C}\setminus\{j\}} \left(1 - \sum_{k\in\mathcal{C}\setminus(\{i\}\cup\{j\})} a_{ki}\right) \mathbf{h}_j^\top \cdot \mathbf{h}_i/\|\mathbf{h}_j\|^2\right.$$

$$\left. - \sum_{k\in\mathcal{V}\setminus\mathcal{C}}\sum_{i\in\mathcal{C}\setminus\{j\}} I_{ik}\mathbf{h}_j^\top \cdot \mathbf{h}_k/\|\mathbf{h}_j\|^2\right) \tag{20}$$

$$= \sum_{i\in\mathcal{C}\setminus\{j\}} \frac{K-2}{n-1}\mathbf{h}_j^\top \cdot \mathbf{h}_i/\|\mathbf{h}_j\|^2 - \sum_{k\in\mathcal{V}\setminus\mathcal{C}} \frac{K-1}{n-1}\mathbf{h}_j^\top \cdot \mathbf{h}_k/\|\mathbf{h}_j\|^2$$

$$\square$$

## C  Theorem 3.3 and Proof

**Theorem 3.3** (Absorbing Probabilities and Node Influence). *Consider the Markov chain with a transition matrix $\tilde{\mathbf{A}}$ derived from a graph $G$. Denote the probability of being absorbed in the $j$-th state (absorbing state) when starting from the $i$-th state as $b_{i,j}$. Then, $I(v_i, v_j) = b_{i,j}$, where $I(v_i, v_j)$ is the node influence from the $j$-th node $v_i$ to the $i$-th node $v_j$ on $G$.*

*Proof.* Following Lemma A.1, we can obtain:

$$
\begin{aligned}
\left\|\frac{\partial \mathbf{h}_i}{\partial \mathbf{h}_j}\right\| = \left\|\frac{\partial \mathbf{h}_j}{\partial \mathbf{h}_j}\right\| \cdot &\left(\left(\frac{1}{D_{ii}} a_{ik_1^1} \frac{1}{D_{k_1^1 k_1^1}} a_{k_1^1 k_2^1} \cdots \frac{1}{D_{k_{n_1}^1 k_{n_1}^1}} a_{k_{n_1}^1 j}\right)\right. \\
&\left.+ \cdots + \left(\frac{1}{D_{ii}} a_{ik_1^m} \frac{1}{D_{k_1^m k_1^m}} a_{k_1^m k_2^m} \cdots \frac{1}{D_{k_{n_m}^m k_{n_m}^m}} a_{k_{n_m}^m j}\right)\right) \\
= &\left(\frac{1}{D_{ii}} a_{ik_1^1} \frac{1}{D_{k_1^1 k_1^1}} a_{k_1^1 k_2^1} \cdots \frac{1}{D_{k_{n_1}^1 k_{n_1}^1}} a_{k_{n_1}^1 j}\right) \\
&+ \cdots + \left(\frac{1}{D_{ii}} a_{ik_1^m} \frac{1}{D_{k_1^m k_1^m}} a_{k_1^m k_2^m} \cdots \frac{1}{D_{k_{n_m}^m k_{n_m}^m}} a_{k_{n_m}^m j}\right).
\end{aligned}
\tag{21}
$$

As illustrated in Lemma A.1, $\|\partial \mathbf{h}_j / \partial \mathbf{h}_j\| = 1$ because $\|\partial \mathbf{h}_j / \partial \mathbf{h}_j\| = \|\mathbf{I}\| = \sup_{\|\mathbf{h}\|=1}\{\|\mathbf{Ih}\|\} = 1$. On the other hand, utilizing the total probability law and the properties of Markov chains, we know

$$
b_{i,j} = \sum_{k=1}^{n} b_{k,j} p_{ik},
$$

where $p_{ik}$ is the $(i,j)$-entry of $\mathbf{P}$. The reason is that assuming the current state is $i$, we know the next state will be $k$ with probability $p_{ik}$. Then we can iteratively expand the expression as follows:

$$
\begin{aligned}
b_{i,j} &= \sum_{k=1}^{n} p_{ik} \sum_{l=1}^{n} p_{kl} \cdots \sum_{o=1}^{n} p_{mo} b_{o,j} \\
&= \left(\left(p_{ik_1^1} p_{k_1^1 k_2^1} \cdots p_{k_{n_1}^1 j} b_{j,j}\right)\right. \\
&\quad \left.+ \cdots + \left(p_{ik_1^m} p_{k_1^m k_2^1} \cdots p_{k_{n_m}^m j} b_{j,j}\right)\right) \\
&= \left(\left(p_{ik_1^1} p_{k_1^1 k_2^1} \cdots p_{k_{n_1}^1 j}\right)\right. \\
&\quad \left.+ \cdots + \left(p_{ik_1^m} p_{k_1^m k_2^1} \cdots p_{k_{n_m}^m j}\right)\right).
\end{aligned}
\tag{22}
$$

In the above equation, we apply the similar expansion idea as in Eq. (7). The result is obtained by accumulating all products of transition probabilities on all possible paths from state $i$ to state $j$. It is noteworthy that there are multiple absorbing states on this Markov chain. However, we ignore the paths absorbed in other absorbing states, since the corresponding absorbing probability $b_{k,j}$ is 0, where $k$ and $j$ are two absorbing states. In this way, we further incorporate $b_{j,j} = 1$ and obtain the final result. Considering Eq. (21) and Eq. (22), we can find that by setting $p_{ik} = a_{ik}/D_{ii}$, we can obtain $\|\partial \mathbf{h}_i / \partial \mathbf{h}_j\| = b_{i,j}$. Moreover, we know $\sum_{k=1}^{n} p_{ik} = \sum_{k=1}^{n} a_{ik}/D_{ii} = 1, i = 1, 2, \ldots, t$. Therefore, these transition probabilities satisfy the requirements of Markove chains, which completes the proof. □

## D  Theorem 3.4 and Proof

**Theorem 3.4** (Approximation of Absorbing Probabilities)**.** *Denote $t$ as the number of non-absorbing states in the Markov chain, i.e., $t = |\mathcal{V}_{\mathcal{T}}| - |\mathcal{S}|$, where $|\mathcal{V}_{\mathcal{T}}|$ is the node set in $G_{\mathcal{T}}$. Denote $\mathbf{Q} \in \mathbb{R}^{t \times t}$ as the transition probability matrix for non-absorbing states in the Markov chain. Estimating $b_{i,j}$ with $\tilde{b}_{i,j} = \sum_{k=1}^{t} \tilde{\mathbf{A}}_{k,j} \cdot \sum_{h=0}^{m} \left(\mathbf{Q}^h\right)_{i,k}$, then the absolute error is upper bounded as $|b_{i,j} - \tilde{b}_{i,j}| \leq C/t^{m-1}(t-1)$, where $m$ controls the upper bound, and $C$ is a constant.*

*Proof.* Considering the transition probability matrix $\mathbf{P}$ as

$$
\mathbf{P} = \begin{pmatrix} \mathbf{Q} & \mathbf{R} \\ \mathbf{0} & \mathbf{I}_r \end{pmatrix},
$$

where $\mathbf{Q} \in \mathbb{R}^{t \times t}$ and $\mathbf{R} \in \mathbb{R}^{t \times r}$. $\mathbf{0}$ is a $t \times r$ zero matrix, and $\mathbf{I}_t$ is an $r \times r$ identity matrix. Basically, the absorbing probability $b_{i,j}$ (i.e., the probability of being absorbed in the $j$-th state when starting from the $i$-th state) can be represented as

$$
\begin{aligned}
b_{i,j} &= \sum_{k=1}^{t} P(X_{t+1} = j | X_t = k) \cdot \mathbb{E}(N(k) | X_0 = i) \\
&= \sum_{k=1}^{t} p_{kj} \cdot \sum_{h=0}^{\infty} \left( \mathbf{Q}^h \right)_{i,k},
\end{aligned}
\tag{23}
$$

where $N(k)$ is the umber of visits to state $k$. Nevertheless, directly calculating $\sum_{h=0}^{\infty} \mathbf{Q}^h$ can be inefficient. Thus, we propose to estimate it by the sum of the first $h$ values. Specifically, we know $\left( \mathbf{Q}^h \right)_{ik}$ is the probability that state $i$ transitions to state $j$ in exactly h steps. Therefore,

$$
\begin{aligned}
\left( \mathbf{Q}^h \right)_{ik} &= \sum_{k_1=1}^{t} p_{ik_1} \sum_{k_2=1}^{t} p_{k_1 k_2} \cdots \sum_{k_h=1}^{t} p_{k_h k} \\
&= \left( \left( p_{ik_1^1} p_{k_1^1 k_2^1} \cdots p_{k_h^1 j} \right) + \cdots + \left( p_{ik_1^m} p_{k_1^m k_2^m} \cdots p_{k_h^m j} \right) \right),
\end{aligned}
$$

which is the sum of transition probability products on all possible paths of length $h$ from state $i$ to state $j$. Furthermore,

$$
\begin{aligned}
\left( \mathbf{Q}^h \right)_{ik} &\leq C * \max \left( \left( p_{ik_1^1} p_{k_1^1 k_2^1} \cdots p_{k_h^1 k} \right) \right. \\
&\quad , \cdots , \left. \left( p_{ik_1^m} p_{k_1^m k_2^m} \cdots p_{k_h^m k} \right) \right) \\
&= C * \left( p_{ik_1^*} p_{k_1^* k_2^*} \cdots p_{k_h^* k} \right) \\
&= C * \left( \sqrt[h]{p_{ik_1^*} p_{k_1^* k_2^*} \cdots p_{k_h^* k}} \right)^h \\
&= C * \left( \mathrm{GM}(p_{ik^*}^{(h)}) \right)^h,
\end{aligned}
\tag{24}
$$

where $C$ is the extracted constant, and $\mathrm{GM}(p_{ik^*}^{(h)})$ denotes the geometric mean of the path with the transition probability products. Based on the inequality of arithmetic and geometric means (i.e., the AM-GM inequality), we know the geometric mean is always less than or equal to the arithmetic mean. Therefore, $\mathrm{GM}(p_{ik^*}^{(h)}) \leq \mathrm{AM}(p_{ik^*}^{(h)}) = 1/t$. This inequality holds because there are totally $t$ non-absorbing states in the Markov chain, and the transition probabilities to all states sum up to 1. As a result, we can obtain:

$$
\left( \mathbf{Q}^h \right)_{ik} \leq C * \left( \mathrm{GM}(p_{ik}^{(h)}) \right)^h \leq C/t^h.
\tag{25}
$$

Recalling Eq. (23), if we only keep the terms with power less than or equal to $m$, the estimation error can be presented as follows:

$$
|b_{i,j} - \tilde{b}_{i,j}| = \sum_{k=1}^{t} p_{kj} \cdot \sum_{h=m+1}^{\infty} \left( \mathbf{Q}^h \right)_{i,k} \leq \sum_{k=1}^{t} p_{kj} \cdot \sum_{h=m+1}^{\infty} C/t^h.
\tag{26}
$$

Since $p_{kj} < 1$ is a transition probability, we know $\sum_{k=1}^{t} p_{kj} < t$. Therefore,

$$
\sum_{k=1}^{t} p_{kj} \cdot \sum_{h=m+1}^{\infty} C/t^h \leq tC \sum_{h=m+1}^{\infty} 1/t^h = tC \cdot \frac{1/t^{m+1}}{1 - 1/t} = \frac{C}{t^{m-1}(t-1)}.
\tag{27}
$$

In this way, we can obtain the final inequality:

$$
|b_{i,j} - \tilde{b}_{i,j}| \leq \sum_{k=1}^{t} p_{kj} \cdot \sum_{h=m+1}^{\infty} C/t^h \leq \frac{C}{t^{m-1}(t-1)}
\tag{28}
$$

$\square$

# E  Details on Experiments

In this section, we introduce the datasets, parameter settings for our framework and baselines, and training and evaluation details.

## E.1  Datasets

In this section, we provide further details on the five datasets used in the experiments. (1) Tissue-PPI [49] consists of 24 protein-protein interaction (PPI) networks from different tissues. The node features are obtained based on gene signatures, and node labels are gene ontology functions [16]. Each label corresponds to a binary classification task, where the total number of such labels is 10. (2) Fold-PPI [18] is a dataset provided by G-Meta, constructed from 144 tissue networks. The labels are assigned based on the corresponding protein structures defined in the SCOP database. Specifically, fold groups with more than nine unique proteins are selected, resulting in 29 unique labels. Node features are conjoint triad protein descriptors [26]. Different from Tissue-PPI where all nodes are assigned labels, the labeled nodes in Fold-PPI are relatively scarce, which better fits into the few-shot scenario. (3) DBLP [30] is a citation network, where each node represents a paper, and links between nodes are created based on the citation relationship. The node features are generated from the paper abstracts, and the class labels denote the paper venues. (4) Cora-full [2] is a citation network with node labels assigned based on the paper topic. This dataset extends the prevalent small dataset via extracting original data from the entire network. (5) ogbn-arxiv [17] is a directed citation network which consists of all CS arXiv papers indexed by MAG [35], where nodes represent arXiv papers, and edges indicate citation relationships. The node labels are assigned based on 40 subject areas of arXiv CS papers.

## E.2  Parameter Settings

In this section, we introduce the detailed parameter settings for our experiments. For the ogbn-arxiv dataset, the number of update steps is 40 with a meta-learning rate of 0.005 and a base learning rate of 0.1. For other single-graph datasets, the number of update steps is 20 with a meta-learning rate of 0.005 and a base learning rate of 0.1. For the Tissue-PPI dataset, the number of update steps is 20 with a meta-learning rate of 0.005 and a base learning rate of 0.01. For the Fold-PPI dataset, the number of update steps is 20 with a meta-learning rate of 0.005 and a base learning rate of 0.1. The hidden dimension sizes of GNNs are set as 16. The dropout rate is set as 0.5. The weight decay rate is set as $10^{-4}$. For the approximation of absorbing probabilities, we set the value of $m$ as 2. For the common sampling, we set the value of $C$ as 10. For the local sampling, we set the value of $h$ as 2 (i.e., 2-hop neighbors). The activation functions are all set as the ReLU function.

## E.3  Baseline Settings

In this section, we introduce the detailed settings for baselines used in the experiments. (1) KNN [13]: We follow the settings in G-Meta to first train a GNN on all training data as an embedding function. During test, we assign a label for each query node based on the voted K-closest node in the support set. (2) ProtoNet [27]: This baseline classifies query nodes based on their distances to the learned prototypes (i.e., the average embedding of support nodes in a specific class). We set the learning rate as 0.005 with a weight decay of 0.0005. (3) MAML [14]: This baseline performs several update steps within each meta-task and meta-updates the parameter based on query loss. The meta-learning rate is set as 0.001 and the number of update steps is 10 with a learning rate of 0.01. (4) Meta-GNN [48]: This baseline combines MAML with Simple Graph Convolution (SGC) [40]. Following the original work, we set the learning rate and meta-learning rate as 0.5 and 0.003, respectively. (5) G-Meta [18]: This baseline leverages the local subgraphs to learn node embeddings while combining ProtoNet and MAML. Following the original work, we set the numbers of update steps for meta-training and meta-test as 10 and 20, respectively. The inner learning rate is 0.01 while the outer learning rate is 0.005. The hidden dimension size is set as 128. (6) GPN [10]: This baseline learns node importance and utilizes the ProtoNet to classify query nodes. We follow the setting in the source code and set the learning rate as 0.005 with a weight decay of 0.0005. The dimension sizes of two GNNs used in GPN are set as 32 and 16, respectively. (7) RALE [21]: This baseline learns node embeddings based on the relative and absolute locations of nodes. We set the learning rates for training and fine-tuning as 0.001 and 0.01, respectively. The hidden size of used GNNs is 32.

### E.4 Details on Training and Evaluation

We train our model on a single 16GB Nvidia V100 GPU. The GNNs used in our experiments are implemented with Pytorch [24], which is under a BSD-style license. The required packages are listed as below.

- Python == 3.7.10
- torch == 1.8.1
- torch-cluster == 1.5.9
- torch-scatter == 2.0.6
- torch-sparse == 0.6.9
- torch-geometric == 1.4.1
- numpy == 1.18.5
- scipy == 1.5.3

For the training and evaluation setting, we adopt different choices for single-graph and multiple-graph settings. Specifically, for the single-graph datasets, we adopt two settings: 5-way 3-shot (i.e., $N = 5$ and $K = 3$) and 10-way 3-shot (i.e., $N = 10$ and $K = 3$). For the multiple-graph datasets, we adopt 3-way 3-shot (i.e., $N = 3$ and $K = 3$) for Fold-PPI and 2-way 5-shot (i.e., $N = 2$ and $K = 5$) for Tissue-PPI. This is because Tissue-PPI consists of 10 binary classification tasks. The number of training epochs is set as 500. Furthermore, to keep consistency, the meta-testing tasks are identical for all baselines. For the class split setting in the single-graph datasets, we set training/validation/test classes as 15/5/20 for ogbn-arxiv, 25/20/25 for Cora-full, and 80/27/30 for DBLP. For the class split setting in the multiple-graph datasets, the class split setting on the disjoint label task is 21/4/4 for Fold-PPI and 8/1/1 for Tissue-PPI (each label in Tissue-PPI corresponds to a binary classification task). The graph split setting is 80%/10%/10%, which follows the same setting as G-Meta.

## F Supplementary Discussion

### F.1 Limitations

This paper aims at learning task-specific structures for each meta-task to promote few-shot node classification performance. Nevertheless, certain disadvantages exist in our specific design. First, the learned task-specific structures is tailored for one meta-task (including the support set and the query set) and cannot be easily generalized to other meta-tasks. In consequence, when a scenario setting requires a significantly larger query set than the support set, the generalization to all these query nodes can be difficult due to the large query set size. Second, when the graph size is relatively small, the original graph can be sufficient for conducting few-shot tasks. As a result, the learned task-specific structures can potentially lead to loss of useful information when the graph size is excessively small.

### F.2 Negative Impacts

This paper studies the problem of graph few-shot learning, which exists widely in real-world applications. For example, certain protein structures maintain scarce labeled proteins, which increases the difficulty of classification on such protein structures [18]. Moreover, the technique is novel and suitable for various scenarios. Therefore, we currently do not foresee any negative impacts in our proposed framework.

### F.3 Potential Improvements

**Preserving the Original Graph.** In our design, we select a specific number of nodes to construct the task-specific structure for each meta-task. Nevertheless, during this process, useful information in other nodes can be lost. Although through our design, we can maintain the majority of useful information in the selected nodes, the nodes that are not selected can still be helpful. Therefore, a potential strategy for improvements can be keeping the original graph while learning flexible edge weights among all the nodes. In this way, the information inside all nodes will be preserved for better performance. This strategy is especially helpful for datasets with small graphs, where all the nodes

can be potentially informative. Nevertheless, when the graph size becomes larger, such a strategy can lead to scalability problems.

**Introducing Multiple GNNs.** Although the learned task-specific structure is tailored for the meta-task, the GNNs are only applied to this structure. As a result, the information propagation process is restricted in the task-specific structure. A possible improvement strategy is to leverage another GNN that propagates messages on the original graph. Meanwhile, the learned features can be incorporated into the task-specific structure. In this way, the incorporated features are task-agnostic and can help the learning in each meta-task in a comprehensive manner. Nevertheless, such a strategy is not suitable for the multiple-graph setting, since learning GNNs across different graphs can lead to suboptimal performance.