# OpenReview forum: "Graph Few-shot Learning with Task-specific Structures"
_NeurIPS.cc/2022/Conference — NeurIPS 2022 Accept_

### Official Review · Reviewer_PD5y · 2022-07-02

**Rating:** 6
**Confidence:** 5
**Soundness:** 3 good
**Presentation:** 2 fair
**Contribution:** 3 good

**Summary:**

The authors have proposed a new mechanism to learn **task-specific** structure for each meta-task under the problem setup of few-shot node representation learning. The intuition behind the idea is clear in that the node representations should be learned in compliance with the structure of the meta-task being considered. As the authors mention, its evident that the entire graph is not necessarily useful since it may contain a substantial amount of redundant information, especially since each meta-task usually has a small number of nodes (this may not be the case when the graph at hand is small, nevertheless, most of the real world are much larger compared to the size considered in the experiments). Furthermore, the authors suggest that the edge weights over the task-specific structure should also be learned in a task-specific manner. The proposed methodology seeks to answer two questions in particular - (i) how to select relevant nodes for the task specific structure and (ii) how to compute the edge weights for the task specific structure. To answer the first question, the authors derive an approximation based on the “influence” of one node embedding to another and give a lower bound to the expected value of this influence across the nodes in a class in terms of the “shortest path distance” (SPD) amongst the nodes. Based on the SPD criterion, the authors have proposed two simple strategies to sample the nodes - local and common sampling. To answer the second question, the authors compute the edge weights as a combination of two different terms obtained using - node representations and SPD between the two given nodes. To further assist the learning process of the task-specific structures, the authors propose 2 loss functions based on the - (i) label information of the support set nodes and (ii) mutual information between the query nodes and their estimated / potential labels. For the first one, a loss function that enforces the maximization of the similarity within classes along with the minimization of the similarity between classes, is proposed (eq 4 of the main paper). For the second one, the maximization of mutual information is suggested. The optimization of the model is performed via MAML. Experiments on multiple popular benchmarks and comparison against state of the art baselines demonstrates that the proposed approach indeed performs very well. Although, there are several concerns regarding the theoretical analysis, as elaborated below.

**Questions:**

1. The authors have explicitly focused on selecting nodes for learning task-specific structures, conditioned on the support set. While this is a perfectly sound method, why haven’t the authors considered the same for the query set nodes? Even if a single node is present in the query set, the same proposed mechanisms might help expand the subgraph around the query set node to tackle bottlenecks, particularly oversquashing, as well as densify the graph in certain cases (on average), to prevent under reaching from the nodes in the set $\mathcal{V_T}$ \ $\mathcal{Q}$. Moreover, the first limitation mentioned by the authors in appendix (lines 177-180) can potentially be addressed via a similar mechanism. It will be interesting to hear the authors’ opinion about this point.
2. What specific GNN architecture has been used to obtain node representations in the proposed model?
3. Can the authors provide more details on how the hyperparameters are selected? Are these searched over some space or selected based on previous works?
4. The improvements suggested in section F.3 (appendix), although simple, can be utilized in cases where the scalability is not an issue. Although the statement - “Nevertheless, such a strategy is not suitable for the multiple-graph setting, since learning GNNs across different graphs can lead to suboptimal performance” - lines 204-206 in appendix, isn’t compelling. Why do the authors mention that learning one, or infact multiple GNNs can be suboptimal? These GNNs can learn the original graph structures and provide useful auxiliary information.


**Limitations:**

I commend the authors for explicitly stating some of the main limitations of the approach. A question regarding the first limitation (lines 177-180) has also been raised above. I strongly suggest the authors to explicitly elaborate the GNN architecture that has been used in the experiments, as there could be potential concerns regarding the architecture used. I have **serious** concerns regarding the theoretical analysis, as mentioned above. Lastly, I don’t see any serious negative societal impacts of the work.


**Strengths And Weaknesses:**

1. The empirical evidence of the efficacy of the proposed method is strong. All the results in tables 2 and 3 indicate that the model outperforms the baselines by substantial margins on both - multi graph and single graph settings.
2. The result of the ablations in figure 2 also explicitly demonstrate the utility of each of the 3 main components of the proposed method. The explanations provided in lines 322-332 of the main paper are also inline with the demonstrated results.
3. The authors haven’t provided sensitivity analyses for many important hyperparameters, such as - the value of $C$ in common sampling, value of $h$ in local sampling, value of $m$ in eq 6 etc. I believe such analyses are important to further interpret the proposed solution better.
4. I believe that the authors should also provide some qualitative results along with the quantitative evidence. For eg - constructing t-SNE (or any other) plots showing the class wise separation over the embeddings of the penultimate layer can be a useful analysis of the learned embeddings. This can be done for the proposed method and the best performing baseline. Similarly, some analysis of the “average size of the task-specific graph structure” will be useful. Additionally, some plots for the “performance-runtime” concerning the proposed method and the best performing baseline will provide an interesting tradeoff analysis.
5. Notational problems in eq 6 of the main paper - $j$ is the class label.
6. Definition 1 (in the main paper) as well as the proof of Lemma A.1 (appendix) have multiple notational problems: (i) The layer indexing for $h_i$ and $h_j$ are missing. From the proof, it's evident that the authors have considered layer recursion upto the shortest path length between $i$ and $j$; (ii) $d$ and $\bar{d}$ are the same. $\bar{d}$ is not defined in the statement of Lemma A.1
7. Eq 6 in the appendix hasn’t used Jensen's inequality properly in the first comparison.
8. Multiple problems in proof of theorem 3.2, following the concerns above about the layer indexing and the entire proof sketch thereof. The assumption that the expected value of each $I_{ij} = \frac{1}{n-1}$ since the sum of probabilities of all paths starting from $v_{i}$ equals 1, is extremely loose.
9. It will be useful to compute the value of the constant $C$ in theorem 3.4, since it seems to follow the order of $t^{h}$ (computing its upper bound might be easier than the actual value)

---

> ### Author Response · Authors · 2022-08-02
> **Response to Official Review**
>
> Thank you for your insightful advice! Hope our response can answer your questions!
>
> **Question 1**: The authors haven’t provided sensitivity analyses
>
> **Answer 1**: Thank you for your suggestions! We provide more analyses regarding different values of hyper-parameters under the 5-way 3-shot setting.
> | Value of C | 1    | 3    | 5    | 10   | 20   |
> |------------|------|------|------|------|------|
> | DBLP       | 76.7 | 76.8 | 79.0 | 79.3 | 78.2 |
> | Cora-full  | 64.3 | 64.2 | 65.7 | 66.2 | 66.3 |
> | ogbn-arxiv | 56.9 | 57.4 | 58.1 | 58.5 | 58.6 |
>
> | Value of h | 1    | 2    | 3    | 4    |
> |------------|------|------|------|------|
> | DBLP       | 77.1 | 79.3 | 79.0 | 78.8 |
> | Cora-full  | 64.2 | 66.2 | 66.1 | 65.6 |
> | ogbn-arxiv | 57.9 | 58.5 | 58.5 | 58.1 |
>
>
> We conduct experiments to present the effect of different values of hyper-parameters C and h in common sampling and local sampling, respectively. Increasing both these hyper-parameters will result in a larger task-specific structure size. From the results, we can observe that the performance is relatively low when C=1 and h=1, which indicates a small task-speicifc structure size. Nevertheless, further increasing these values does not generally provide improvements and can even degrade the performance. This is because although a larger size can potentially learn more beneficial task-specific structures, it can also incorporate irrelevant information that reduces the performance. Moreover, a larger size will inevitably lead to a higher computational cost, while the performance improvement is relatively small.
>
> **Question 2**: Notational problems in eq 6
>
> **Answer 2**: Here j is the class label, which corresponds to the j-th absorbing state in the Markov Chain. And this state corresponds to the j-th node on the learned task-specific structure, which is the support node.
>
> **Question 3**: Layer indexing in Definition 1
>
> **Answer 3**: We omit the layer index because we assume that the number of layers is large enough to express the graph. The proof is based on the output of GNNs, which is irrelevant to the number of layers.
>
>
> **Question 4**: The assumption in Theorem 3.2 is loose.
>
> **Answer 4**: The expectation is typically reasonable in practice. In this theorem, we consider the influence from a node to all other nodes of its class on the entire graph. Therefore, considering that the class set size on each graph is relatively small, the number of such nodes is generally large enough to hold the expectation assumption.
>
>
> **Question 5**: Why not consider the same for the query set nodes
>
> **Answer 5**: The reason is that the size of the query set can vary in realistic scenarios. If the size is only one, the result cannot be guaranteed. Also, if the classification result of a query node depends on other query nodes, the performance cannot be fairly evaluated. Thus, we do not use the query set for the task-specific structure and ensure that the framework is applicable in various sizes of query sets. Nevertheless, in a test scenario with a large query set, we can also construct the task-specific structure for better generalization performance. We believe your idea is insightful and beneficial in certain scenarios, where utilizing an entire query set can provide benefits. In the future, we can further investigate this direction to explore the advantages.
>
>
> **Question 6**: What specific GNN architecture has been used to obtain node representations in the proposed model?
>
> **Answer 6**: We use GCNs as our base models for fair comparisons with other baselines. Since our framework focuses on the task-specific structure, it can fit into a wide range of GNNs.
>
>
> **Question 7**: How are the hyper-parameter selected?
>
> **Answer 7**: For specific hyper-parameters that are commonly used in previous works (e.g., the number of GNN layers and the hidden size), we generally keep them the same as baselines. For other hyper-parameters that are proposed in our work, we manually search over a range to select them (specifically, [1,3,5,10,20] for C in common sampling, [1,2,3,4] for h in local sampling, and [1,2,3,4] for m in Eq. 6).
>
>
> **Question 8**: Why do the authors mention that learning one, or in fact, multiple GNNs can be suboptimal?
>
> **Answer 8**: Our concern is that under the multiple-graph few-shot scenario, the optimization of GNNs will be more difficult. This is because the meta-tasks on different graphs are distinct, which could result in different optimization objectives. As a result, although the learned GNNs can be used for multiple graphs, their generalizability across meta-tasks is not guaranteed.  Nevertheless, such a design can also be tested with multiple graphs to empirically evaluate its performance.

---

> > ### Comment · Reviewer_PD5y · 2022-08-05
> > **Response to Rebuttal**
> >
> > I commend the authors for providing additional experiments and responding to the questions asked.
> >
> > While the concerns surrounding the experiments have been mostly addressed, I have the following concerns regarding the theoretical results:
> > The layer indexing matters because the influence values are calculated conditioned on this. Thereby, all the results following "Definition 1" hold across the layers and not within the given layer to which node $h^{l}_{i}$ ( $l$ being the layer) belongs (for eg - the work of [1]).
> >
> > Although interestingly, the proposed objectives make sense intuitively and the results clearly demonstrate the efficacy of the work, thus I don't see any major concern in the experiments. I am thus increasing the score to **6**.
> >
> > [1] Understanding over-squashing and bottlenecks on graphs via curvature , ICLR 2022

---

> > > ### Author Response · Authors · 2022-08-06
> > > **Response to Response**
> > >
> > > Thank you for your advice! Indeed, the influence can be impacted by layer indexing. Nevertheless, after enough layers of propagations, the influence can be represented by the outputs of GNNs and is irrelevant to specific layers.
> > >
> > > Thank you so much! We will keep up the good work!

---

### Official Review · Reviewer_SEHi · 2022-07-11

**Rating:** 6
**Confidence:** 2
**Soundness:** 3 good
**Presentation:** 2 fair
**Contribution:** 3 good

**Summary:**

This paper proposes an improved MAML algorithm that tailors node representation to specific meta-tasks. The result is a significant performance improvement across meta-tasks.

**Questions:**

How does the node influence in the paper relate to the field of graph influence maximilation? It seems that the definition, stripping the GNNs from it, is quite similar to the one of Kempe et al., 2003.

What is the computational complexity of the distance computations, during training and for a new task?

**Limitations:**

Limitations and societal impact are addressed in the paper.


**Strengths And Weaknesses:**

[S] The problem is well-motivated and relevant to the research community. The proposed algorithm seems interesting and novel, while the approach intuitive.

[S] The performance improvements observed are significant compared to the baselines.

[W] The terminology is rather hard to follow in the beginning, especially before reading the introduction to the problem.

(comment) The attached code is extremely counterintuitive and messy, even for research code. I had a hard time trying to clarify some computational aspects of the paper for myself.

---

> ### Author Response · Authors · 2022-08-02
> **Response to Official Review**
>
> Thank you for your insightful advice! Hope our response can answer your questions!
>
> **Question 1**: The terminology is hard to follow in the beginning.
>
> **Answer 1**: Thank you for your advice! We have modified the original paper and increased the clarity in line 30-36.
>
> **Question 2**: How does the node influence in the paper relate to the field of graph influence maximization?
>
> **Answer 2**: The definitions of these two types of influence are different in two aspects. First, the graph influence focuses on the influence of certain nodes towards the entire graph, while the node influence measures the impact from a specific node to another node. Second, the node influence is based on the output of GNNs, while the definition of graph influence does not include GNNs. Thus, although the terms are similar for node influence and graph influence, the inner concepts are different.
>
> **Question 3**: What is the computational complexity of the distance computations, during training and for a new task?
>
> **Answer 3**: The time complexity of the commonly used Dijkstra algorithm is O((V+E)logV), where V and E are the numbers of nodes and edges, respectively. Moreover, the training of single-graph settings is conducted on a graph, which means the shortest distances can be calculated and stored before training. Therefore, the computation cost during training can be further reduced.

---

> > ### Comment · Reviewer_SEHi · 2022-08-08
> > **acknowledged**
> >
> > My score stays the same – it seems that the paper is reasonably solid.
> > My only recommendation is to put the runtime complexity in the paper (adjusted to the number of source nodes you run the computation from).

---

> > > ### Author Response · Authors · 2022-08-09
> > > **Response to Reviewer**
> > >
> > > Dear Reviewer,
> > >
> > > Thank you so much! We really appreciate your suggestions and would like to include the analysis in the paper. We will keep up the good work!
> > >
> > > Thank you!

---

> ### Author Response · Authors · 2022-08-07
> **Looking Forward to Your Feedback**
>
> Dear Reviewer,
>
> Thank you for your advice! We have responded to your questions and concerns.
>
> We are willing to answer any further questions. Looking forward to your feedback!
>
> Thank you!

---

### Official Review · Reviewer_AUxb · 2022-07-11

**Rating:** 6
**Confidence:** 3
**Soundness:** 3 good
**Presentation:** 3 good
**Contribution:** 3 good

**Summary:**

This paper proposes a method to select relevant nodes and learn desirable edge weights for each few-shot node classification task, so that the input graph structure is tailored for each task individually. Specifically, the method has three stages: 1) selecting nodes based on proximity to the support set nodes; 2) optimizing edge weights function such that the resulting node influence between support nodes in the same class is maximized; 3) optimizing edge weights function such that the mutual information n between query nodes and all labels in the task is maximized. Over the produced graphs, the authors then use a meta-optimization method based on MAML to learn and perform the few-shot tasks. Experiment results over diverse settings demonstrate the effectiveness of the method over various baseline methods.

**Questions:**

- Is the training of edge weight function using the two objectives completed before the meta-optimization? Or how are they blended together?
- It would be beneficial to confirm the performance gains with statistical tests due to the high variance.
- What is computational overhead with this method over using MAML on the full graphs directly?

**Limitations:**

The limitations are adequately addressed in the paper other than the ones mentioned above.

**Strengths And Weaknesses:**

# Strengths
- The proposed method is novel and well-motivated from practical and theoretical perspectives. However, it is worth noting that similar ideas have been explored and somewhat standardized in the link prediction literature following SEAL (https://proceedings.neurips.cc/paper/2018/file/53f0d7c537d99b3824f0f99d62ea2428-Paper.pdf), where a specific subgraph is selected for predicting the existence of an edge even in the standard (non few-shot) setting. Yet to my knowledge, the most popular methods of this kind simply select an enclosing neighborhood of the two end nodes based on simple heuristics (e.g. intersection of k hop neighborhoods). Therefore, this work is still more novel and sophisticated in the specific methods.
- The method can serve as a simple drop-in preprocessing step for various other methods, and it can potentially benefit tasks even outside few-shot setting or node classification setting.
- The experiment is comprehensive and demonstrates empirical gains.
- The paper is clearly written overall.

# Weaknesses
- The experiment results have significant variance, making it less clear how significant the results are. It would be beneficial to confirm the gains with statistical tests.
- It seems computationally intensive to precompute and extract subgraphs for each task.

---

> ### Author Response · Authors · 2022-08-02
> **Response to Official Review**
>
> Thank you for your insightful advice! Hope our response can answer your questions!
>
> **Question 1**: The experiment results have significant variance.
>
> **Answer 1**: Although we observed variance during the test, this is a common phenomenon in few-shot node classification tasks. This is potentially due to the randomness of sampling meta-tasks in few-shot learning. Since each meta-task maintains a different class set, the classification result can vary among meta-tasks. Therefore, the results are averaged over 500 meta-test tasks for a fair comparison. In fact, most baselines maintain a similar magnitude of variance.
>
> **Question 2**: It seems computationally intensive to precompute and extract subgraphs for each task.
>
> **Answer 2**: The computation cost is not intensive in our framework. If we directly extract subgraphs from the original graph, the computation cost can be large. Instead, our framework first selects relevant nodes and then learns the edge weights among these selected nodes, which results in a task-specific structure. Since the number of selected nodes is relatively small, the computation cost can be well controlled.
>
> **Question 3**:  Is the training of edge weight function using the two objectives completed before the meta-optimization? Or how are they blended together?
>
> **Answer 3**: We simultaneously learn the task-specific edge weights and conduct the meta-optimization. They are optimized with different losses, which is described in Section 3.4.
>
> **Question 4**: What is the computational overhead with this method over using MAML on the full graphs directly?
>
> **Answer 4**: The computation cost of our framework is controllable. If MAML is combined with GATs or other structure learning methods on the full graphs, the process will be conducted on the entire graph. Therefore, the graph size is the entire graph, which can be intensive when the full graph is large. Instead, our framework selects a small number of relevant nodes from the full graph for the task-specific structure. As a result, the size of the task-specific structure is significantly smaller than the full graph, which largely reduces the computation cost.

---

> > ### Comment · Reviewer_AUxb · 2022-08-07
> > **Thank you for the response**
> >
> > The response addressed most of my concerns and questions. It would be good to include some of the discussions and clarifications to the paper eventually. I will raise my score to 6.

---

> > > ### Author Response · Authors · 2022-08-08
> > > **Thank you!**
> > >
> > > Dear Reviewer,
> > >
> > > Thank you so much! We appreciate the discussions and would like to include some in the paper. We will keep up the good work!

---

> ### Author Response · Authors · 2022-08-07
> **Looking Forward to Your Feedback**
>
> Dear Reviewer,
>
> Thank you for your advice! We have responded to your questions and concerns.
>
> We are willing to answer any further questions. Looking forward to your feedback!
>
> Thank you!

---

### Official Review · Reviewer_zmmr · 2022-07-14

**Rating:** 5
**Confidence:** 4
**Soundness:** 1 poor
**Presentation:** 2 fair
**Contribution:** 2 fair

**Summary:**

Considering graph few-shot learning, the paper proposes a method called GLITTER to learn node representations across different meta-tasks.
In particular, GLITTER selects nodes according to their overall node influence on support nodes, learns edge weights in the task-specific structure based on node influence within classes and mutual information between query nodes and labels. The authors believe that these make the learned structures tasks-specific and consequently improve the node representations.



**Questions:**

The key problem is that the authors fails to convince the readers why existing methods cannot learn task-specific structures. The authors should provide a clear definition for their believed task-specific structure.

Indeed, in meta-learning, after the inner loop fine-tuning, the model and the node embeddings naturally become task-specific. In this sense, existing meta-learning works can learn task-specific structures.
On line 5-6, the authors write that "these methods generally rely on the original graph (i.e., the graph that the meta-task is sampled from) to learn node representations". On line 39, the authors write that "Here each meta-task is often sampled from one single graph".
Then, why this problem exists in multiple-graph setting? If the problem is that one should adjust the provided graph adaptively, why not using GAT [1] or recent graph structure learning methods [2-5]? Any difference or relevance w.r.t them?

[1] Graph attention networks, ICLR, 2018

[2] Few-shot learning with graph neural networks, ICLR, 2018

[3] Learning to propagate labels: Transductive propagation network for few-shot learning, ICLR, 2018

[4] Property-Aware Relation Networks for Few-Shot Molecular Property Prediction, NeurIPS, 2021

[5] Deep graph structure learning for robust representations: A survey, 2021



**Limitations:**

The authors did not discuss any limitations nor potential negative societal impact of their work.

**Strengths And Weaknesses:**

The paper considers broader graph few-shot learning applications following GMeta.
The results show clear improvement.

However, this paper is confusing due to unclear motivation and claim.
Please see my detailed comments in questions.

---

> ### Author Response · Authors · 2022-08-02
> **Response to Official Review**
>
> Thank you for your insightful advice! We would like to clarify that our motivation is reasonable and distinct from most previous studies. We also provide plentiful theoretical analyses in the paper to support our claims. Hope our response can answer your questions!
>
> **Question 1**: The authors fail to convince the readers why existing methods cannot learn task-specific structures.
>
> **Answer 1**: The existing methods (e.g., meta-learning) cannot learn task-specific structures. Although the meta-learning strategy can learn task-specific embeddings for nodes and graphs via fine-tuning, it can only be used on the entire graph with GNNs, which inevitably increases the difficulty of extracting task-specific structural information. In other words, the graph structures are not “task-specific” during fine-tuning. Instead, our framework learns task-specific edge weights for each meta-task based on node influence and mutual information. In this way, the learned task-specific structure can promote the learning of task-specific embeddings.
>
>
> **Question 2**: Why not use GATs or other structure learning methods?
>
> **Answer 2**: The reason is that GATs or other structure learning methods can only learn the task-specific structure for the entire graph, since they do not have a node selection strategy. As a result, the irrelevant nodes can intervene in the fine-tuning process. In contrast, we propose to select nodes via two strategies according to their overall node influence on support nodes in each meta-task, which can effectively reduce the adverse impact of irrelevant nodes.
>
>
> **Question 3**: Why the problem of sampling from one graph exists in multiple-graph settings?
>
> **Answer 3**: This is because in the multiple-graph settings, the node classification in the test scenario is conducted only on one novel graph. Therefore, during training, each meta-task is solely sampled from one graph to keep consistency. To increase clarity, We have modified the original paper and provided an explanation in line 39-40.
>
>
> **Limitation 1**: The authors did not discuss any limitations nor potential negative societal impact of their work.
>
> **Answer**: We discussed the limitations and negative societal impacts of our framework in Appendix F. Specifically, our framework can be less effective when the query set size is significantly larger than the support set. Moreover, when the graph size is relatively small, the original graph is sufficient for few-shot learning without the need for task-specific structures.

---

> ### Author Response · Authors · 2022-08-07
> **Looking Forward to Your Feedback**
>
> Dear Reviewer,
>
> Thank you for your advice! We have responded to your questions and concerns.
>
> We are willing to answer any further questions. Looking forward to your feedback!
>
> Thank you!

---

> ### Author Response · Authors · 2022-08-08
> **Appreciating Your Feedback**
>
> Dear Reviewer,
>
> Thank you so much for your advice. We hope that our answers can address your concerns.
>
> We are looking forward to your feedback!
>
> Thank you!

---

> > ### Comment · Reviewer_zmmr · 2022-08-09
> > **Thanks for the reply.**
> >
> > Thanks for the reply. Most of my concerns are solved.
> > I now increase my final score to five.

---

> > > ### Author Response · Authors · 2022-08-09
> > > **Response to Reviewer**
> > >
> > > Dear Reviewer,
> > >
> > > Thank you so much! We really appreciate your cognition of our paper. We will keep up the good work!
> > >
> > > Thank you!

---

### Meta-Review · Area_Chair_cE9P · 2022-08-30

**Recommendation:** Accept
**Confidence:** Less certain

**Metareview:**

The paper attempts to improve few shot learning over graphs. In this regards, the authors propose a multi-stage approach where first relevant nodes is identified and desirable edge weights is learnt for each few-shot node classification task, so that the input graph structure is tailored for each task individually. This proposed method is based on insights from theoretical analysis. Experiments are carried to show the proposed method is more effective over some baseline methods. We thank the authors and reviewers for actively engaging in discussion and taking steps towards improving the paper including for providing additional experiments. Some concerns about theoretical analysis remain on interaction of influence between nodes and layers and would be nice to discuss in the final version.

Other minor fixes:
- Line 23 in appendix typo: fix partial latex symbol
- Eq 6 below line 38: First equality should be inequality

**Award:**

No

---

### Decision · Program_Chairs · 2022-09-14

Accept